# Neolithic *Yersinia pestis* infections in humans and a dog
Julian Susat[1], Magdalena Haller-Caskie [1], Joanna H. Bonczarowska [1,2], Nicolas A. da Silva [1], Kerstin Schierhold [3], Michael M. Rind[3], Ulrich Schmölcke [4], Wiebke Kirleis[5], Holger Sondermann[6,7], Christoph Rinne[5], Johannes Müller [5], Almut Nebel[1] & Ben Krause-Kyora [1] ✉

*Yersinia pestis* has been infecting humans since the Late Neolithic (LN). Whether those early infections were isolated zoonoses or initiators of a pandemic remains unclear. We report *Y. pestis* infections in two individuals (of 133) from the LN necropolis at Warburg (Germany, 5300–4900 cal BP). Our analyses show that the two genomes belong to distinct strains and reflect independent infection events. All LN genomes known today (*n* = 4) are basal in the phylogeny and represent separate lineages that probably originated in different animal hosts. In the LN, an opening of the landscape resulted in the introduction of new rodent species, which may have acted as *Y. pestis* reservoirs. Coincidentally, the number of dogs increased, possibly leading to *Y. pestis* infections in canines. Indeed, we detect *Y. pestis* in an LN dog. Collectively, our data suggest that *Y. pestis* frequently entered human settlements at the time without causing significant outbreaks.

Plague is considered one of the most notorious scourges of humanity and was responsible for at least three pandemics in historical times[1]. Its causative agent, *Yersinia pestis*, has been infecting humans for more than 5000 years. The oldest *Y. pestis* genomes so far have been detected in remains of a hunter-gatherer from Riņņukalns, Latvia (RV2039; 5300–5050 cal BP)[2] and a farmer from Gökhem, Sweden (Gökhem_2; 5040–4867 cal BP)[3] (Fig. 1A), here referred to as Late Neolithic (LN). The two LN genomes represent strains in the early stages of *Y. pestis* evolution and were shown to be ancestral to a phylogenetically distinct lineage termed Late Neolithic and Bronze Age (LNBA; 4800–2500 cal BP), which was present throughout Eurasia until at least the third millennium BP[4]. Both LN strains lacked the virulence factors and mutations observed in much later forms that were required for efficient transmission from rodents to humans via fleas[2–5]. However, it remains unclear whether the *Y. pestis* infections detected in the LN skeletal remains were due to isolated zoonoses or marked the beginning of a long-lasting pandemic across Eurasia sustained by human-to-human contact[2–4,6]. Another question centers on what may have facilitated the transmission of *Y. pestis* to humans before adaptations to the flea evolved. Various as yet unknown rodent species could have acted as primary hosts for *Y. pestis* at that time. Human exposure to these rodents and possibly *Y. pestis* might have been increased through domesticated carnivores, i.e. dogs, which hunted these animals. This could also have led to *Y. pestis* infections in the canines themselves. In this study, we extend our understanding of

*Y. pestis* evolution during the LN by presenting two new genomes from human individuals and providing evidence of a *Y. pestis* infection in a dog.

## Results

Here, we report on two new LN *Y. pestis* genomes that were identified in human remains from the Warburg necropolis (5300–4900 cal BP), located in present-day Germany (Fig. 1A). The site is archaeologically attributed to the LN Wartberg Culture (WBC)[7] and consists of five gallery graves (I-V)[7] with commingled skeletal remains of at least 198 individuals[8] (Fig. 1B). For 133 individuals from gallery graves I, III and IV, we generated genome-wide shotgun sequences. Metagenomic screening of the datasets for *Y. pestis* DNA revealed two positive individuals, hereafter labelled Warburg_1 (5291-4979 cal BP; from gallery grave I) and Warburg_2 (5265-4857 cal BP; from gallery grave III). Their remains were discovered in two separate gallery graves and the radiocarbon dates likely placed them in different times. With a likelihood of 88%, Warburg_1 is older than Warburg_2 and the probability distributions indicate a difference of about 200 years (OxCal v4.4.4., Supplementary Data 1, Supplementary Fig. 1). We found that the two individuals had typical WBC ancestry components[9] and were not related to each other (Supplementary Data 2 and 3, Supplementary Figs. 2 and 3). Regarding *Y. pestis* reads, both samples showed a high mean coverage of up to 28 x for the chromosome and 117 x for the three plasmids (Table 1, Supplementary Data 1, Supplementary Figs. 4 and 5). The genomes

[1]Institute of Clinical Molecular Biology, Kiel University, Kiel, Germany. [2]Department of Genetics, Yale School of Medicine, New Haven, CT, USA. [3]LWL-Archäologie für Westfalen, Münster, Germany. [4]Centre for Baltic and Scandinavian Archaeology (ZBSA), Schloss Gottorf, Schleswig, Germany. [5]Institute of Pre- and Proto-historic Archaeology, Kiel University, Kiel, Germany. [6]Centre for Structural Systems Biology (CSSB), Deutsches Elektronen-Synchrotron DESY, Hamburg, Germany. [7]Kiel University, Kiel, Germany. ✉e-mail: b.krause-kyora@ikmb.uni-kiel.de

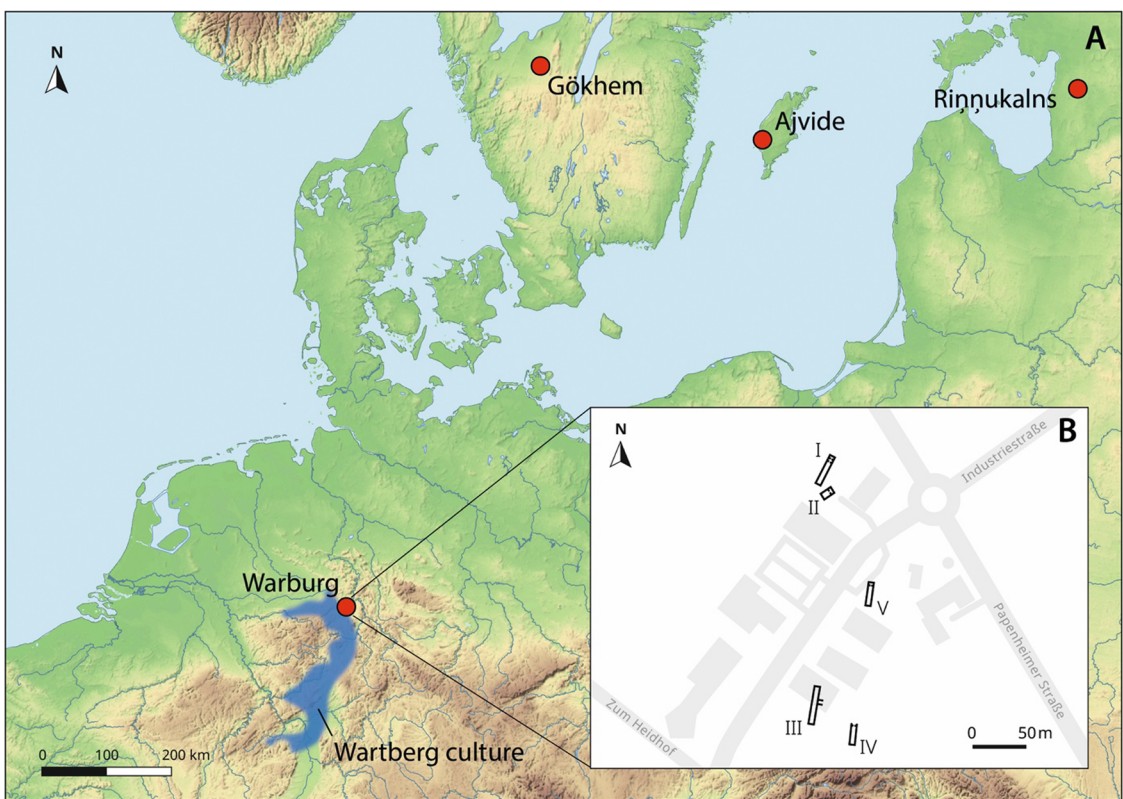

**Fig. 1 | Map of LN sites.** correctLocation of archaeological sites in which LN *Y. pestis* was identified in human (Warburg, Riṇṇukalns, Gökhem) and canine remains (Ajvide) (**A**). Warburg necropolis with the gallery graves I, III-V and building II (**B**).

had virulence factors described for modern *Y. pestis*, apart from the *ymt* gene, the filamentous prophage YpfΦ and YPMT1.66c that are all known to be absent in LN strains[4,5,10] (Supplementary Fig. 6). Consistent with this finding, all four LN genomes had a basal position in the phylogenetic tree (Figs. 2 and 3). Warburg_1 branched off shortly after RV2039 and Warburg_2 diverged shortly before Gökhem_2. To assess the potential influence of single-nucleotide polymorphisms (SNPs) characteristic of the LN lineage (Warburg_1, Warburg_2, RV2039, Gökhem_2), we carried out a SNP effect analysis. We focused only on non-synonymous SNPs which lead to nonsense or missense mutations due to the loss/gain of a start/stop codon. In doing so, we discovered six SNPs in two members of the LN branch that were not shared by later-dating strains (Supplementary Data 4). These six SNPs either introduce or remove stop codons. The introduction of a stop codon in these cases creates short N-terminal protein fragments, unlikely to retain the function of the protein. Removal of a stop codon leads to C-terminal extensions by translation of intergenic regions or potential fusion proteins including sequences from the neighbouring gene. Based on the sequences, a functional assessment is not possible, and additional experiments are required to distinguish between the two possibilities. The accumulation of pseudogenes, which is a hallmark of the LNBA clade[4], was not visible in the LN genomes. This may indicate that adaptation to one specific host had not yet occurred.

To test the hypothesis that the prehistoric dog could have been infected with *Y. pestis*, we screened publicly available shotgun datasets from Eurasian Neolithic (*n* = 15) and Bronze Age canines (*n* = 6)[11–15] for pathogens in general and *Y. pestis* in particular. In one canine (labelled C90), we detected *Y. pestis*-specific reads which had not been reported in the original publication[12]. Specimen C90 consisted of a mandible found in the cultural layers of the settlement site Ajvide on the island of Gotland in present-day Sweden and dated to 4900–4500 cal BP (Pitted Ware Culture; Fig. 1A)[12]. The reads in C90 showed the typical damage profiles for ancient DNA. Although the data was previously only generated for canine population genetic analyses[12], we could reconstruct 8% of the bacterial genome. However, due

to the low coverage, we did not explore its placement in the phylogenetic tree.

## Discussion

In this study, we identified *Y. pestis* in two individuals from the LN necropolis at Warburg in Germany. Our results suggest that the two *Y. pestis* genomes (Warburg_1 and Warburg_2) belonged to distinct strains. They differed in 82 positions (Supplementary Fig. 7). Due to this genetic distance, it can be assumed that the two genomes did not directly evolve from each other. This finding is also reflected in the typology of the phylogenetic tree indicating a divergence of the genomes long before the infections occurred (Fig. 3). In addition, the genomes were detected in unrelated individuals who were buried in different gallery graves. All evidence suggests that both infections represent independent events and thus appear not epidemiologically linked.

Interestingly, despite the screening of numerous specimens from each gallery grave (total *n* = 133), we found no further infections with *Y. pestis* or any other pathogen. In another collective WBC grave at the site of Niedertiefenbach (*n* = 42; present-day Germany), no signs of pathogens were detected either[9]. It must be acknowledged that pathogen-negative results do not necessarily mean absence of infection, as taphonomic processes may have degraded any microbial traces. In addition, both the Warburg and Niedertiefenbach samples consisted of petrous bones as well as teeth, the latter being the better source material for the detection of blood-borne viruses and bacteria[16]. These limitations notwithstanding, the arguments presented above (i.e., independent infection events at Warburg) and the overall small number of 2 positives among the 175 tested WBC individuals suggest that the collective graves were not used for the burial of victims of a plague outbreak or other epidemics, as previously suggested for the same period[3]. The few *Y. pestis* cases in the WBC are consistent with the results of other large-scale pathogen screenings that have so far revealed only single infections with human pathogens (*Y. pestis*, *Salmonella enterica*) or endemically occurring infections (hepatitis B virus, parvovirus B19, *Helicobacter*

## Table 1 | Mapping statistics for the *Y. pestis*-positive samples

| Sample | Mapping reads | Cov 1x (%) | Cov 2x (%) | Cov 3x (%) | Mean coverage |
|---|---|---|---|---|---|
| **chromosome NC_003143.1** | | | | | |
| Warburg_1 | 521,157 | 89.45 | 76.32 | 58.09 | 3.22 x |
| Warburg_2 | 3,315,529 | 94.84 | 94.68 | 94.58 | 28.89 x |
| C90 | 31,549 | 8.42 | | | 0.31 x |
| **pCD1 NC_003131.1** | | | | | |
| Warburg_1 | 16,491 | 92.14 | 90.62 | 88.44 | 8.38 x |
| Warburg_2 | 94,944 | 92.86 | 92.33 | 92.09 | 43.96 x |
| C90 | 263 | 12.45 | | | 0.14 x |
| **pMT1 NC_003134.1** | | | | | |
| Warburg_1 | 7,857 | 57.52 | 33.57 | 16.49 | 1.5 x |
| Warburg_2 | 75,513 | 74.55 | 73.04 | 72.67 | 17.03 x |
| C90 | 187 | 6.04 | | | 0.07 x |
| **pPCP1 NC_003132.1** | | | | | |
| Warburg_1 | 4,386 | 79.85 | 79.80 | 79.67 | 11.34 x |
| Warburg_2 | 42,311 | 80.28 | 80.20 | 80.19 | 117.31 x |
| C90 | 63 | 20.92 | | | 0.26 x |

Overview of statistics after mapping to the chromosome and plasmids of the *Y. pestis* reference genome. The abbreviation "cov" refers to the percentage of the genome with at least 1 x, 2 x or 3 x coverage, respectively.

*pylori*) in Neolithic remains[17]. Also, the mortuary practice of single and multiple inhumations during that period does not indicate mass mortality, as would be expected in an epidemic. The findings from the Neolithic are thus in marked contrast to the short-term mass burials and the high pathogen load seen in the Middle Ages[18,19].

The two Warburg genomes increase the number of *Y. pestis* genomes from the LN to four. All LN genomes were distinct from each other and represent lineages separated by an extended period of independent evolution (Figs. 2 and 3). The high diversity and the basal position of the LN *Y. pestis* lineages may reflect a low level of specialization at this early evolutionary stage. Such reduced specialization potentially facilitated their survival across diverse environments and a wide host range. According to the current phylogeny, the LN strains gave rise to two lineages, one from which the pathogens of the deadly Justinianic and medieval plagues emerged and another that led to the LNBAs (Fig. 2). The LNBA clade went extinct sometime in the third millennium BP[4] (Figs. 2 and 3). For more than 2000 years, the LNBA strains were the dominant *Y. pestis* lineage in humans across Eurasia[4]. They may represent an adaptation to a very specialized *Y. pestis* ecology (e.g., host(s)), as reflected by the increasing pseudogenization of bacterial genes over time[4]. This process could have led to the evolutionary dead-end of the LNBA lineage and to a less severe, perhaps even chronic, manifestation of plague in humans that resembled an endemic rather than a pandemic disease[2,4].

During the LN, woodland clearance increasingly created open landscapes in central and northern Europe[20–23] that attracted a variety of new rodent species (e.g., European hamster *Cricetus cricetus*[24]) originally native to the steppe further east or south. Some of these species could have been natural reservoirs of *Y. pestis*[1] and an infection in humans would have been feasible through close contact with a *Y. pestis*-positive wild animal or carcass[25,26]. However, we do not know how frequent such encounters were, especially when the animals in question were not normally hunted or touched by humans. Therefore, we propose the dog as facilitator which could have increased exposure of humans to *Y. pestis* from various wild animals, especially those with which humans did not come into regular contact. The archaeological record during the LN shows large numbers of dog remains, for instance, as material for ornamentation (pendants and jewellery made from dog teeth, e.g., 356 teeth [canines] in the grave Wewelsburg 118 in Altendorf)[27]. In addition, the animals were likely used for hunting and

herding[28,29]. If dogs preyed on infected animals, this could have increased the probability of *Y. pestis* transmission from rodents to humans. Since modern dogs can develop pneumonic plague and infect humans directly without the need for flea adaptation[1,26], the question arises whether this was also the case during the LN. Given instances of dog-to-human transmissions today[1], it is possible that dogs themselves acted as a *Y. pestis* reservoir for humans at the time, or vice versa.

To the best of our knowledge, C90 is so far the only case of *Y. pestis* infection in a Neolithic dog. This small number may be explained by the innate resistance of dogs to *Y. pestis* which leads to rapid pathogen clearance and a low fatality rate[1,30]. If Neolithic dogs recovered from plague as frequently and quickly as their modern counterparts[30], most that were ever infected would be *Y. pestis*-negative at the time of their death (and in the palaeogenomic pathogen screening). Thus, we might underestimate the actual number of infected dogs.

The presence of *Y. pestis* in a dog from present-day Sweden fits well with the geographical distribution pattern of the four infected human individuals (Fig. 1A). Surprisingly, all five *Y. pestis* findings from the LN occupy a relatively small geographical area in northwestern Europe. Overall, the results suggest a significant *Y. pestis* exposure in and around human settlements at the time, most likely leading to isolated infections rather than large-scale disease outbreaks.

## Methods

### Human skeletal samples

Archaeologically, the Warburg necropolis belongs to the Late Neolithic Wartberg culture (WBC) and is dated to 5300–4900 cal BP. The burial complex is located 1.6 km northwest of the town Warburg (Germany). It consisted of four stone gallery graves (I, III, IV, V) and one wooden burial chamber or ritual building (II)[7]. The gallery graves contained commingled skeletal remains of at least 198 individuals[8]. In total, we included bone and tooth samples from 133 individuals in the pathogen screening for which we could generate at least 2.5 million sequencing reads each.

### DNA isolation, sequencing and processing

All lab work was carried out in the Kiel Ancient DNA Laboratory, following established guidelines[31]. Tooth and bone samples were cleaned with pure bleach, rinsed with purified water, dried overnight at 37 °C, and subsequently ground in a ball mill homogenizer for 45 sec at maximum speed. Between 80–120 mg of powder was used for extraction based on a published protocol[32] as outlined in a previous study[33]. The double-stranded half-UDG libraries with unique index combinations for each sample were shotgun sequenced on an Illumina HiSeq platform (2x100bp) and the generated data was pre-processed by removing sequencing adaptor remnants and merging overlapping reads with ClipAndMerge v1.7.8[34].

### Metagenomic screening

For the metagenomic screening, a custom database was created with the "MALT-build" function of the software MALT[35], following the manual's instructions. For this purpose, all bacterial and viral genomes that included the description "complete genome" were downloaded from NCBI[36] (as of 12.04.2021). This final database comprised 38,273 complete bacterial and viral genomes. Subsequently, metagenomic screening on both the pre-processed human (*n* = 133) and published canine datasets (*n* = 21)[11–15] was performed using the "MALT-run" function in semi-global alignment mode with a 90% sequence identity threshold. The output files were visualized in MEGAN6[37].

### *Yersinia pestis* alignment

For the two human *Y. pestis*-positive samples Warburg_1 and Warburg_2, more shotgun data was generated from the initial libraries without additional enrichment. Subsequently, the reads of the three *Y. pestis*-positive samples (human: Warburg_1 and Warburg_2; canine: C90[12]) were mapped against the reference genome of the *Y. pestis* CO92 chromosome (NC_003143.1) and the three plasmids pCD1 (NC_003131.1), pMT1

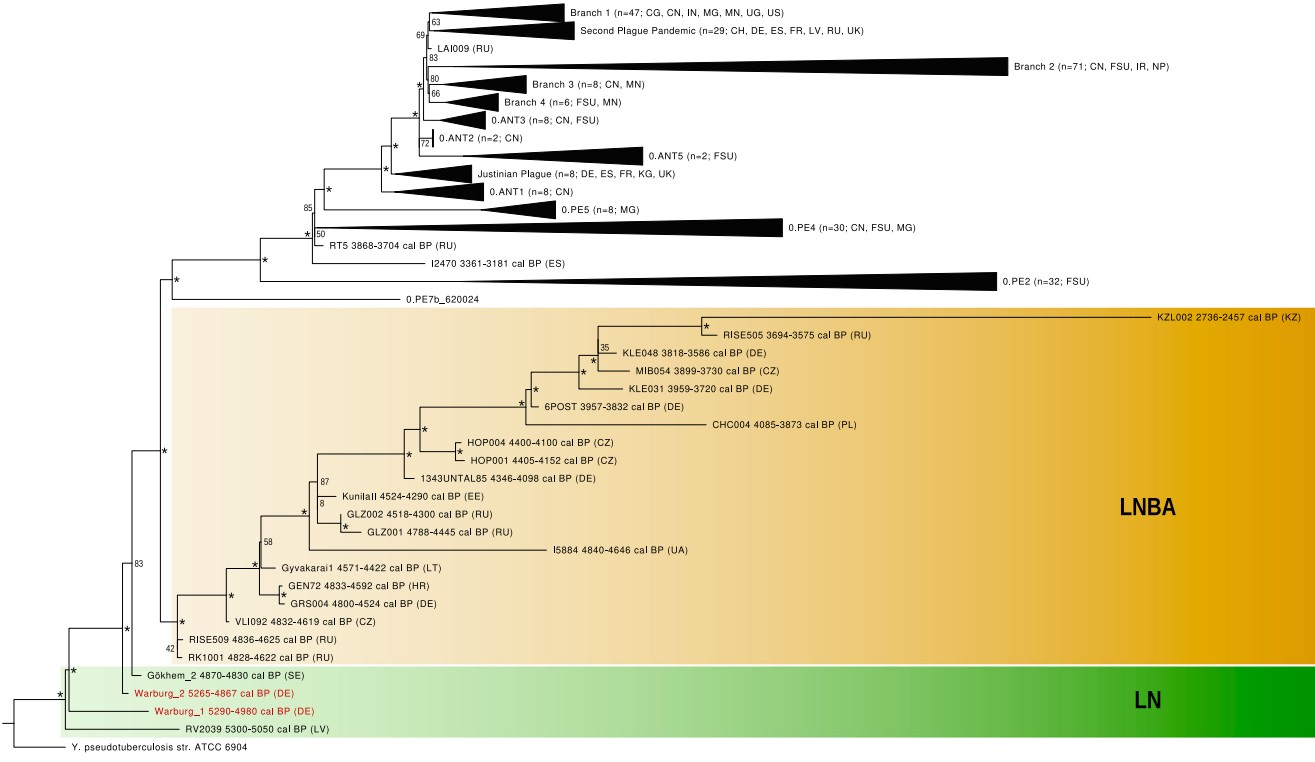

**Fig. 2 | Phylogeny of ancient and modern *Y. pestis* genomes.** Maximum likelihood phylogenetic tree based on the SNP alignment (10,743 positions) of 226 modern genomes, 62 published ancient genomes, the novel genomes Warburg_1 and Warburg_2 (red) and the outgroup *Y. pseudotuberculosis*. LN strains are highlighted in green and LNBA strains in yellow. Dating of the ancient strains is given as calibrated years before the present (cal BP). Country abbreviation is given in brackets (CH = Switzerland, CG = Congo, CN = China, CZ = Czech Republic, DE = Germany, EE = Estonia, ES = Spain, FR = France, FSU = Former Soviet Union, HR = Croatia, IN = India, IR = Iran, KG = Kyrgyzstan, KZ = Kazakhstan, LT = Lithuania, LV = Latvia, MG = Madagascar, MM = Myanmar, MN = Mongolia, NP = Nepal, PL = Poland, RU = Russia, SE = Sweden, UA = Ukraine, UG = Uganda, UK = United Kingdom, US = United States). Unique positions to the outgroup were excluded to facilitate the visualisation. Bootstrap values were calculated for 1000 replicates and nodes with a support above 90 are marked with an asterisk. The scale corresponds to substitutions per site. Genomes included in the phylogeny are listed in Supplementary Data 5.

(NC_003134.1) and pPCP1 (NC_003132.1). The ancient origin of the samples was authenticated with DamageProfiler v1.1 (Supplementary Data 1, Supplementary Figs. 4 and 5)[38].

## SNP effect analysis

VCF files from Warburg_1 and Warburg_2 were filtered for SNPs with a coverage of at least 3 x, a calling quality of 30 and a 90% majority call. This set of filtered SNPs was provided to SNPEff v4.3[39] to analyze the effects of SNPs in the ancient genomes. The output file was filtered for high-impact SNPs and evaluated. A SNP effect analysis was not possible on the data from the dog genome (C90) due to insufficient coverage.

## Analysis of virulence factors

The *Y. pestis* reads from Warburg_1 and Warburg_2 were screened for the presence or absence of 115 chromosomal and 44 plasmid-associated virulence genes compiling a bed file based on supplementary information of a previous study[4]. To asses the coverage of these virulence genes for each *Y. pestis* genome, the percentage of the coverage was determined using bedtools v2.25.0[40] with the "genomecov" and "coverage" functions. A short awk script (awk '$5! = 0 {print $1 "\t" $4 "\t" $8}' [sampleID].histogram.bed > [sampleID].coverage_table.bed) was used to extract columns for chromosome/plasmid name ($1), gene name ($4) and gene coverage ($8) from the bed file. The values were all collected in an excel table that was used as input for R v3.6.3[41]. Heatmaps were created with the R pheatmap package.

## *Yersinia pestis* phylogenetic analysis

Generation of VCF files and phylogenetic analyses were based on 226 modern and 62 ancient strains, including Warburg_1, Warburg_2 (Supplementary Data 5), and carried out as described in Susat et al.[42]. To ensure the authenticity of the SNPs used, we took the following precautions. For a SNP to be called, it must be supported by 90% of the reads and a minimum coverage of 4 reads. SNPs located in previously identified homoplastic regions[4] were excluded (based on a list provided by https://github.com/aidaanva/LNBAplague/blob/main/multiVCFAnalyzer/ SNPstoExclude). For single-stranded libraries generated from samples of the LNBA lineage (downloaded from the open repository ENA, project no. PRJEB51099; libraries KLE031, KLE048), we implemented the publicly available genoSL.R script (https://github.com/aidaanva/GenoSL) to exclude SNPs that are based on the C → T deaminations only visible in single-stranded libraries[4]. Lastly, after generation of the multi-vcf alignment, we excluded *Y. pseudotuberculosis*-specific SNPs to allow easier visualization of the tree, with an intentionally shortened branch for the outgroup *Y. pseudotuberculosis*. Due to insufficient *Y. pestis* coverage in sample C90, a reliable placement of the bacterial genome in the phylogeny was not possible.

## Molecular dating

A SNP-based phylogenetic tree was generated on a subset of 42 representative *Y. pestis* genomes, including Warburg_1 and Warburg_2 (Supplementary Data 6), as described in the previous paragraph. The obtained tree was used as a starting tree for BEAST2 v2.6[43]. Dates were provided in BP (before the present) format, with 1950 AD corresponding to age 0 and the mean age calculated for each dating range. The following parameters were used: uncorrelated relaxed clock, lognormal distribution, GTR + G4 substitution model and a coalescent constant population, as applied in previous studies[2,3,43]. Two different runs were performed: one run

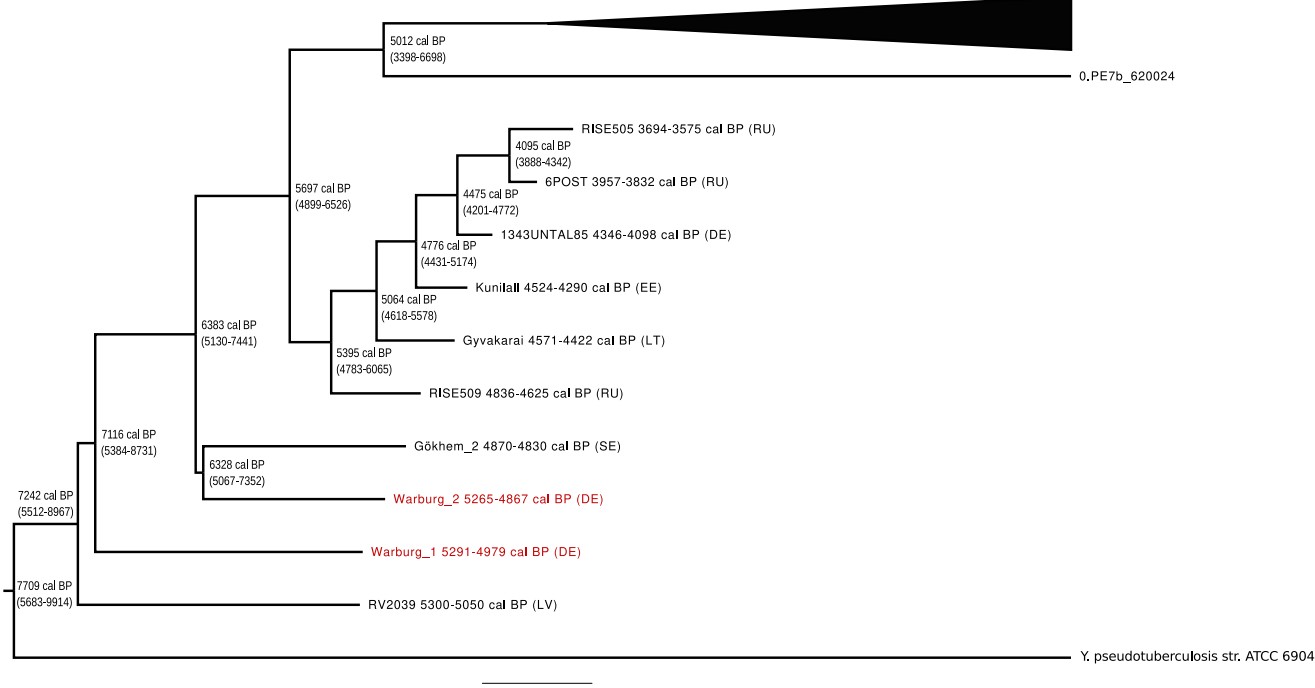

**Fig. 3 | Molecular dating of early *Y. pestis*.** Maximum clade credibility tree based on 42 modern and ancient *Y. pestis* genomes and the two genomes from Warburg (red). All dates are given as calibrated years before the present (cal BP). Country abbreviation is provided in brackets as described in Fig. 2. Genomes included in the molecular dating analysis are listed in Supplementary Data 6.

based on a provided phylogenetic tree that can be altered by the tool, and one run without a starting tree. The multiple output files from both independent runs were combined using the LogCombiner v2.6.7[44] and evaluated with Tracer v1.7.2[45] with effective sample size (ESS) exceeding a minimum value of 250. Trees were combined with TreeAnnotator v2.6.0[44].

### Human population genetic analyses

Reads from Warburg_1 and Warburg_2 were aligned to the human genome reference (hg19). The ancient origin of the DNA was authenticated with DamageProfiler v1.1 (Supplementary Data 1)[38]. Mapping of reads, contamination estimation, genetic sex determination as well as mitochondrial DNA and Y chromosome haplogroup determination were performed as described in Immel et al.[9]. For the two individuals, pseudo-haploid genotypes on ~1240 K informative SNPs[46] were generated and subsequently merged with previously published genotypes from various ancient and modern West Eurasian populations (Allen Ancient DNA Resource)[47]. For the principal component analysis (PCA) with smartpca[48], the genotypes of all ancient populations were projected onto the principal components calculated from the modern populations. An unsupervised ADMIXTURE v1.3.0[49] analysis was performed using 3–12 components (K) with 10 bootstraps each. Additionally, two-way admixture models were tested with qpAdm from Admixtools[50]. Kinship between the two Warburg individuals was assessed with READ[51] using the normalization value of 0.2493 derived from the median mismatch rate observed in the Warburg dataset, which comprised 8256 pairwise comparisons from 129 samples.

### Reporting summary

Further information on research design is available in the Nature Portfolio Reporting Summary linked to this article.

### Data availability

The generated datasets are available in the European Nucleotide Archive under the accession number PRJEB49648.

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

## Acknowledgements

This study was funded by the Deutsche Forschungsgemeinschaft (DFG, German Research Foundation) through the CRC 1266 – project number 290391021.

## Author contributions

A.N. and B.K.-K. conceived and designed the study. K.S. and M.M.R. provided the human skeletal material. J.S., M.H.-C., J.H.B and N.A.d.S. generated and analyzed aDNA data. K.S., M.M.R., U.S., W.K., C.R. and J.M. provided environmental, archaeological and archaeozoological context and interpretation. H.S. interpreted the results of the SNP effect analysis. A.N. and B.K.-K. interpreted the results and wrote the manuscript with input from all the other authors. J.H.B edited the final manuscript.

## Funding

## Competing interests

The authors declare no competing interests.

## Additional information

Materials availability:This study did not generate new unique reagents.

