## [Peer Review File · Communications Biology]

Reviewers' comments:

Reviewer #1 (Remarks to the Author):

In general, the paper fits appropriately with Communications Biology. The authors have performed the necessary bioinformatic analyses needed to support their arguments, and they are offering an exciting insight into the past biology and putative zoonotic transmission path of *Y. pestis* during the Late Neolithic period.

I would highly recommend the authors expand their results section (in the main text) by including more results from their analyses. More specifically, it would strengthen the message of the paper if they included more results regarding the C90 ancient canine sample, as a means of testing their hypothesis for zoonotic transmission. I would also encourage the authors to elaborate on their SNP effect analyses results too.

The results presented along with the conclusions that are drawn definitely expand the scope of pathogen palaeogenomics and highlight the zoonotic potential of *Y. pestis*. I highly recommend that this paper is published in Communications Biology.

This is a list of my review comments:

1. Lines 131-136. This section describes novel results, and as such it could be moved to the 'Results' section, where the authors can expand on their analyses too.
2. Elaborate on the metagenomic screening of publicly available ancient canine genomes the authors performed in the results section. Similarly do so for their SNP effect analyses.
3. As DNA degradation leads to an increased frequency of C to T transitions, which can lead to false positive SNPs being called, thus confounding phylogenetic and population genetic analyses, I would encourage the authors to include a maximum likelihood phylogenetic tree (such as the one in Figure 2) but reconstructed only from transversion SNPs. This would further corroborate their claims and ensure that the phylogenetic placement of the Warburg samples is not due to noise.
4. It would be advisable the authors acknowledged the innate resistance that dogs have to *Yersinia* spp. (Barbieri et al 2020, Nichols et al 2014) when they describe their novel route of transmission for *pestis*, from animals to humans, as this aspect of dog immunity means that their proposed route of transmission would not be that frequent. That could also explain why they did not find more *Y. pestis* in other publicly available ancient canine sequence data.
5. It would be advisable if the authors also mentioned that carriage of *pestis* in dogs does not automatically qualify them as a reservoir/disease vectors, but it does provide a solid basis for future research to discover if they truly ever were one.

Reviewer #2 (Remarks to the Author):

In this article, Susat and colleagues present two new *Y. pestis* genomes derived from Late Neolithic

(LN) human remains. They find that these genomes are not closely related to each other, the two other LN genomes previously published, nor the lineages that led to pandemic plagues or the extinct Late Neolithic/Bronze Age clade of *Y. pestis*. Building on this observation the authors develop the notion that dogs were involved in *Y. pestis* epidemiology at the time, and screen publicly available short read archives from dog archeological remains and identify one individual as being infected by the bacterium.

The article is based on interesting data/findings and should be published. However, it has a weird structure. The dog idea is only brought up in the discussion where all corresponding results are presented: this should be built in from the introduction on and properly presented in the results section. The notion is also brought forth partly based on the unconvincing argument that other animal hosts in the vicinity of humans could not be involved in transmission to humans (which is untrue). Finally, the authors clearly overinterpret their data when they say they show that dogs were a reservoir of *Y. pestis*; the use of the term reservoir implies a directionality (in the case dog to humans) which their data cannot by definition demonstrate.

I would therefore recommend that the authors commit to a major revision of the article.

L28: one of the examples of overinterpretation (or misrepresentation of what the term reservoir implies).

L62: Date ranges are mostly overlapping so the authors should provide more information about the dating here (e.g. that <5% of the date density functions overlapped), rather than simply pointing at sup material.

L66: Please specify the exact coverage in the main text.

L77: This should be mentioned in the results, and the number of snps should be clearly mentioned rather than being vaguely deducible from a heatmap.

L78-81: I think that the authors claim here is that the events are not epidemiologically related, which is true in the literal sense (it was not a direct transmission between these individuals because they were not contemporaneous) and also considering their potential belonging to a very long multi-century transmission chain because of the large genetic distance between the genomes (that diverged from each other about 2,000y before these infections (fig 3)). I think this paragraph could be rewritten to make this point in a clearer manner.

L104: It would make sense to be more specific here and make use of Figure 3 (currently appearing only as an afterthought); e.g. "and represented lineages separated by an extended period of independent evolution (min=1,000 years since divergence of Warburg 2 and Goekhem 2)."

L122-5: I find the argument unconvincing. First, it is totally compatible with the observation of unlinked cases (several independent zoonotic emergence that did not result in large outbreaks). The second part of the argument is very unclear to me: the aforementioned exemplary hosts all had

very broad geographical distributions in Europe a few thousand years ago... More generally, the route of transmission that the authors seem to consider as almost impossibly unlikely is a pretty standard route of transmission of pathogens with similar apparently more or less rare spillovers in humans (hantaviruses in Eurasia or the Americas, Lassa virus in West Africa, monkeypox virus in Central Africa, other orthopoxviruses in many regions of the world, etc.).

L128-9: This is a much stronger argument.

L131-42: This should all go in the results section.

L150-2: The two clauses of this sentence are incompatible, and rely on an improper use of the word reservoir. Generally the conclusion is way too general, and assertive - essentially what can be said is that *Y. pestis* at the time could also, like today, infect dogs. The two new LN genomes and this finding are already interesting pieces of information.

L206: Please describe the methods here, rather than directing at a former article.

Rebuttal letter

We cordially thank the reviewers for their time and comments to improve our manuscript entitled “Neolithic *Yersinia pestis* infections in humans and a curious finding in a dog” (previously “Neolithic humans and dogs – transient reservoirs for *Yersinia pestis*”).

We have carefully revised the manuscript and adapted the structure in line with the reviewers’ recommendations. We have also expanded on the Results, Methods and Discussion sections. We appreciate the reviewers’ comments and have incorporated all their suggestions. Consequently, the manuscript has been substantially improved.

As advised by reviewer 1, we have included more information on the SNP effect analysis. For the interpretation of the results, we have approached Prof. Holger Sondermann for an expert opinion and have added him as a co-author.

In the following, we are addressing each comment separately and indicate which parts of the manuscript have been revised and where these edits can be found in the main text (page and lines).

Reviewers' comments:

Reviewer #1 (Remarks to the Author):

In general, the paper fits appropriately with Communications Biology. The authors have performed the necessary bioinformatic analyses needed to support their arguments, and they are offering an exciting insight into the past biology and putative zoonotic transmission path of *Y. pestis* during the Late Neolithic period.

I would highly recommend the authors expand their results section (in the main text) by including more results from their analyses. More specifically, it would strengthen the message of the paper if they included more results regarding the C90 ancient canine sample, as a means of testing their hypothesis for zoonotic transmission. I would also encourage the authors to elaborate on their SNP effect analyses results too.

The results presented along with the conclusions that are drawn definitely expand the scope of pathogen palaeogenomics and highlight the zoonotic potential of *Y. pestis*. I highly recommend that this paper is published in Communications Biology.

This is a list of my review comments:

1. Lines 131-136. This section describes novel results, and as such it could be moved to the 'Results' section, where the authors can expand on their analyses too.

As suggested, we have moved this part from the Discussion to the Results section (p. 5, lines 90-100). Furthermore, we have also mentioned our hypothesis of a possible canine route of transmission in the Introduction (p. 3, lines 51-56).

2. Elaborate on the metagenomic screening of publicly available ancient canine genomes the authors performed in the results section. Similarly do so for their SNP effect analyses.

We have added a paragraph on the metagenomic pathogen screening of publicly available ancient canine genomes in both the Methods (p. 10, lines 202-207) and Results sections (p. 5,

lines 90-100). We have expanded on the SNP effect analyses in the Methods (p. 11, lines 218-223) and the Results (p. 4, line 77- p.5, line 88).

3. As DNA degradation leads to an increased frequency of C to T transitions, which can lead to false positive SNPs being called, thus confounding phylogenetic and population genetic analyses, I would encourage the authors to include a maximum likelihood phylogenetic tree (such as the one in Figure 2) but reconstructed only from transversion SNPs. This would further corroborate their claims and ensure that the phylogenetic placement of the Warburg samples is not due to noise.

In our first version of the manuscript, we have indeed excluded all false-positive SNPs based on C-to-T transitions which are characteristic of ancient DNA. The maximum likelihood phylogeny presented in the manuscript (Figure 2) is based on transversion SNPs. However, previously we failed to explain this part properly in the Methods section. We have revised this section now and elaborated on our analyses in more detail (p. 12, lines 232-246).

4. It would be advisable the authors acknowledged the innate resistance that dogs have to *Yersinia* spp. (Barbieri et al 2020, Nichols et al 2014) when they describe their novel route of transmission for *pestis*, from animals to humans, as this aspect of dog immunity means that their proposed route of transmission would not be that frequent. That could also explain why they did not find more *Y. pestis* in other publicly available ancient canine sequence data.

Thank you for this comment. We have included this aspect and the references in our text (p. 8, line 170 and 171).

5. It would be advisable if the authors also mentioned that carriage of *pestis* in dogs does not automatically qualify them as a reservoir/disease vectors, but it does provide a solid basis for future research to discover if they truly ever were one.

In line with suggestions from Reviewer #2, we have softened our interpretation of the results and have changed the title of the manuscript accordingly. In addition, we have added the

following sentence: “It should be noted that the presence of *Y. pestis* reads in a dog dated to the 5th millennium BP does not automatically qualify the animal as a disease vector for humans at the time. Nor does it give any indication of the transmission direction.” (p. 9, lines 175-177). We also refer to future research to investigate to which extent prehistoric dogs and other domesticated carnivores were involved in the transmission of *Y. pestis* to humans (p. 9, lines 181-184).

Reviewer #2 (Remarks to the Author):

In this article, Susat and colleagues present two new *Y. pestis* genomes derived from Late Neolithic (LN) human remains. They find that these genomes are not closely related to each other, the two other LN genomes previously published, nor the lineages that led to pandemic plagues or the extinct Late Neolithic/Bronze Age clade of *Y. pestis*. Buildign on this observation the authors develop the notion that dogs were involved in *Y. pestis* epidemiology at the time, and screen publicly available short read archives from dog archeological remains and identify one individual as being infected by the bacterium.

The article is based on interesting data/findings and should be published. However, it has a weird structure. The dog idea is only brought up in the discussion where all corresponding results are presented: this should be built in form the introduction on and properly presented in the results section. The notion is also brought forth partly based on the unconvincing argument that other animal hosts in the vicinity of humans could not be involved in transmission to humans (which is untrue). Finally, the authors clearly overinterpret their data when they say they show that dogs were a reservoir of *Y. pestis*; the use of the term reservoir implies a directionality (in the case dog to humans) which their data cannot by definition demonstrate.

I would therefore recommend that the authors commit to a major revision of the article.

L28: one of the examples of overinterpretation (or misrepresentation of what the term reservoir implies).

We have deleted the term 'reservoir' and softened the interpretation of our results throughout the manuscript, including the Abstract. In addition, we have changed the title of the manuscript.

L62: Date ranges are mostly overlapping so the authors should provide more information about the dating here (e.g. that <5% of the date density functions overlapped), rather than simply pointing at sup material.

We have added more information and have included the following sentence: "With a likelihood of 88% Warburg_1 is older than Warburg_2 and the probability distributions indicate a difference of about 200 years (OxCal v4.4.4., Supplementary Table 1, Supplementary Fig. 1)." (p. 4, lines 66-69).

L66: Please specify the exact coverage in the main text.

We have rephrased the sentence (p. 4, lines 70-73) and added a table with all coverage values to the main text (Table 1).

L77: This should be mentioned in the results, and the number of snps should be clearly mentioned rather than being vaguely deducible from a heatmap.

We have corrected the sentence and stated an absolute number of SNPs (p. 6, line 104).

L78-81: I think that the authors claim here is that the events are not epidemiologically related, which is true in the literal sense (it was not a direct transmission between these individuals because they were not contemporaneous) and also considering their potential belonging to a very long multi-century transmission chain because of the large genetic distance between the genomes (that diverged from each other about 2,000y before these infections (fig 3)). I think this paragraph could be rewritten to make this point in a clearer manner.

Thank you for the advice. We followed the suggestion and added a few more sentences in this regard, also emphasizing Figure 3 (p. 6, lines 104-110).

L104: It would make sense to be more specific here and make use of Figure 3 (currently appearing only as an afterthought); e.g "and represented lineages separated by an extended period of independent evolution (min=1,000 years since divergence of Warburg 2 and Goekhem 2)."

Thank you for the comment. We have incorporated the suggested sentence into the text (p. 7, lines 132-134).

L122-5: I find the argument unconvincing. First, it is totally compatible with the observation of unlinked cases (several independent zoonotic emergence that did not result in large outbreaks). The second part of the argument is very unclear to me: the aforementioned exemplary hosts all had very broad geographical distributions in Europe a few thousand years ago... More generally, the route of transmission that the authors seem to consider as almost impossibly unlikely is a pretty std route of of transmission of pathogens with similar apparently more or less rare spillovers in humans (hantaviruses in Eurasia or the Americas, Lassa virus in West Africa, monkeypox virus in Central Africa, other orthopoxviruses in many regions of the world, etc.).

We have rephrased our arguments in the manuscript (p. 8, lines 153-164).

L131-42: This should all go in the results section.

As also suggested by Reviewer #1, we have moved this part from the Discussion to the Results section (p. 5, lines 90-100).

L150-2: The two clauses of this sentence are incompatible, and rely on an improper use of the word reservoir. Generally the conclusion is way too general, and assertive - essentially what can be said is that *Y. pestis* at the time could also, like today, infect dogs. The two new LN genomes and this finding are already interesting pieces of information.

We have deleted the term 'reservoir' and softened the interpretation of our results (p. 9, lines 175-184).

L206: Please describe the methods here, rather than directing at a former article.

Overall, we have revised the Methods and described the tools used and parameters in more detail, including the molecular dating.

REVIEWERS' COMMENTS:

Reviewer #1 (Remarks to the Author):

The authors have done an excellent job at treating all the comments I previously made. I have no further concerns. The manuscript has improved greatly, and its message is now much clearer and stronger.

Great job and I am looking forward to seeing this published !

Reviewer #2 (Remarks to the Author):

First, my apologies for the delays in providing this review.

The manuscript has much improved and I find most of the answers provided by the authors are acceptable.

That said, I still find that the authors overinterpret their data in places. They sure acknowledge in the final paragraph of the discussion that their data is actually no proof of dogs acting as reservoirs of *Y. pestis* during this period, but the paper is still peppered with wording suggesting they found such evidence.

Here is a full list of textual changes that I deem necessary (not necessarily with my phrasing but with similar meaning):

- Title: rather than "and a curious finding in a dog"; "and a dog"
- Abstract: the second need to be rewritten. It is odd that the authors mention birds specifically and I would remove them. Birds are not believed to play any important role in plague epi nowadays. So maybe something like this: "The LN in northwestern Europe is characterized by an opening of the landscape that resulted in the introduction of new rodent species, which may have acted as *Y. pestis* reservoirs. Coincidentally, the number of dogs increased in settlements, which may have resulted in their occasional infection with *Y. pestis*. Here, we also identify at least one case of LN dog infection with *Y. pestis*. Collectively, these data suggest that *Y. pestis* frequently entered LN human settlements, with no evidence of it causing significant outbreaks."
- L49 and later: "vector" is not used properly. Fleas are a vector for *Y. pestis*, rats are a reservoir. Please replace all occurrences of vector that do not describe fleas with reservoir (if known or reasonably suspected to be the source of the infection of another animal species, or host).
- L51/52: to my knowledge, the role of dogs in the epidemiology of *Y. pestis* in recent times has been absolutely minimal, with extremely rare cases of dog-to-human transmission. Today, dogs have an influence on the epidemiology in humans in 2 ways: 1. By supporting flea populations (but since fleas were likely not vectoring *Y. pestis* during the LN, it does not play a role here), 2. By their contact with and hunting of rodents (which might help flea-based transmission cycles, but could also increase exposure of humans to rodent carcasses, I guess). All in all, their putative role should be better explained here, including the fact that in the LN this role might have been even more anecdotal than what it is today. Again, in my view, what the data demonstrates is not a very plausible zoonotic route, but how commonly *Y. pestis* entered human settlements.
- L55: remove "providing evidence of a putative zoonotic transmission route through the dog" and replace with "one dog". What is documented here is only that LN dogs were sometimes infected with *Y. pestis*. Whether they were involved in the epi of human cases is unknown.
- L68: "a MEDIAN difference"?
- L96: remove "1x"

- L97-100: this sentence would be better placed L94 reight after "... the original publication."
- L104: rather: "They differed at 82 positions."
- L175-184: This final paragraph should be reworked. I insist: the bird/rodent to human through dog route is not parsimonious, and does not align well with what we know of *Y. pestis* today (and even less so when we take into account the few things we know about LN *Y. pestis*). The core message should not be dog-to-human transmission, but the striking fact that boh humans and dogs were infected at the time, pointing at significant exposure in or around human settlements.